

# Dendrobium alkaloids prevent A$\beta_{25-35}$-induced neuronal and synaptic loss via promoting neurotrophic factors expression in mice

Jing Nie[1,2], Yong Tian[2], Yu Zhang[2], Yan-Liu Lu[2], Li-Sheng Li[2] and Jing-Shan Shi[1,2]

[1] Shanghai University of Traditional Chinese Medicine, Shanghai, China
[2] Department of Pharmacology and the Key Laboratory of Basic Pharmacology of Guizhou Province, Zunyi Medical College, Zunyi, Guizhou Province, China

## ABSTRACT

**Background**. Neuronal and synaptic loss is the most important risk factor for cognitive impairment. Inhibiting neuronal apoptosis and preventing synaptic loss are promising therapeutic approaches for Alzheimer's disease (AD). In this study, we investigate the protective effects of Dendrobium alkaloids (DNLA), a Chinese medicinal herb extract, on $\beta$-amyloid peptide segment 25–35 (A$\beta_{25\text{-}35}$)-induced neuron and synaptic loss in mice.

**Method**. A$\beta_{25-35}$(10 μg) was injected into the bilateral ventricles of male mice followed by an oral administration of DNLA (40 mg/kg) for 19 days. The Morris water maze was used for evaluating the ability of spatial learning and memory function of mice. The morphological changes were examined via H&E staining and Nissl staining. TUNEL staining was used to check the neuronal apoptosis. The ultrastructure changes of neurons were observed under electron microscope. Western blot was used to evaluate the protein expression levels of ciliary neurotrophic factor (CNTF), glial cell line-derived neurotrophic factor (GDNF), and brain-derived neurotrophic factor (BDNF) in the hippocampus and cortex.

**Results**. DNLA significantly attenuated A$\beta_{25-35}$-induced spatial learning and memory impairments in mice. DNLA prevented A$\beta_{25-35}$-induced neuronal loss in the hippocampus and cortex, increased the number of Nissl bodies, improved the ultrastructural injury of neurons and increased the number of synapses in neurons. Furthermore, DNLA increased the protein expression of neurotrophic factors BDNF, CNTF and GDNF in the hippocampus and cortex.

**Conclusions**. DNLA can prevent neuronal apoptosis and synaptic loss. This effect is mediated at least in part via increasing the expression of BDNF, GDNF and CNTF in the hippocampus and cortex; improving A$\beta$-induced spatial learning and memory impairment in mice.

Corresponding author
Jing-Shan Shi, zmcshijs@163.com

## INTRODUCTION

Alzheimer's disease (AD) is the most common type of dementia, and is characterized by progressive memory impairment and cognitive decline (*Roberson & Mucke, 2006*). Mounting evidence has indicated that dementia attributed to synaptic dysfunction and neuronal loss in the hippocampus and its associated cortex (*Youssef et al., 2008*; *Selkoe, 2002*; *Niikura et al., 2002*; *Scheff et al., 2007*), which are caused by the accumulation of soluble A$\beta$ oligomers (*Rowan et al., 2007*; *Lacor et al., 2004*; *Haass & Selkoe, 2007*). A$\beta$ is the major constituent in senile plaques and cerebral amyloid angiopathy, which are two most distinctive histopathologies in AD. The accumulation of A$\beta$ in the brain initiates a cascade of events, such as activating astrocytes and microglia, initiating inflammatory responses, which lead to oxidative injury, altering neuronal ionic homeostasis, kinases/phosphatase activities, and so on (*Klafki et al., 2006*). These cascades result in a wide range of neuronal/synaptic dysfunction and loss, as well as loss of neurotrophin retrograde transport, therefore causing patients to present with the symptoms of dementia. Adjustment of the pathological progress of amyloid peptide is a key strategy to slow down the AD progression. In addition to reduction of the A$\beta$ levels in the brain (*Klafki et al., 2006*), the concomitant application of neuroprotective agents may be the alternative therapeutics for AD, and a strategy to prevent progressive synaptic and cognitive degeneration (*Klafki et al., 2006*). Neurotrophins (NT) are synthesized and secreted by the target tissue, and after binding to its receptors and transported in a retrograde manner to the cell body. They exert a wide range of actions, including neuronal survival and differentiation, modulation of neuronal excitability, development and maintenance of synapses and modification of synaptic structure and function (*Poo, 2001*). The application of neurotrophic factors such as BDNF and CNTF could enable the modulation of neuronal survival and synaptic connectivity (*Lu, Christian & Lu, 2008*; *Garcia et al., 2010*)

*Dendrobium nobile* is a traditional Chinese herbal medicine. In our previous studies, alkaloids extract from *Dendrobium nobile* Lindl. (DNLA) showed neuro-protective activity. For example, DNLA can prevent neuronal damages induced by LPS (*Li et al., 2011*; *Zhang et al., 2011*), and oxygen-glucose deprivation and reperfusion (*Wang et al., 2010*), decrease neuronal apoptosis, hyperphosphorylation of tau protein (*Yang et al., 2014*) and A$\beta$ deposition in rat brain (*Chen et al., 2008*). The present study aimed to explore the effects of DNLA in protecting neurons from A$\beta_{25-35}$-induced neurotoxicity in mice, and analyzed the mechanism from the aspects of promoting the secretion of neurotrophic factors.

## MATERIALS AND METHODS

### Drugs and reagents

Dendrobium was collected from Dendrobium planting regions of Xintian Traditional Chinese Medicine Industry Development co., LTD of Guizhou Province in 2014. The dried stems of the herb (10 kg) were extracted by 95% ethanol solution. DNLA was isolated from the extracts, and analyzed by LC-MS/MS. Alkaloids accounted for 79.8% of DNLA, and mainly contained Dendrobine ($C_{16}H_{25}O_2N$, 92.6%), Dendrobine-N-oxide ($C_{16}H_{25}O_3N$,

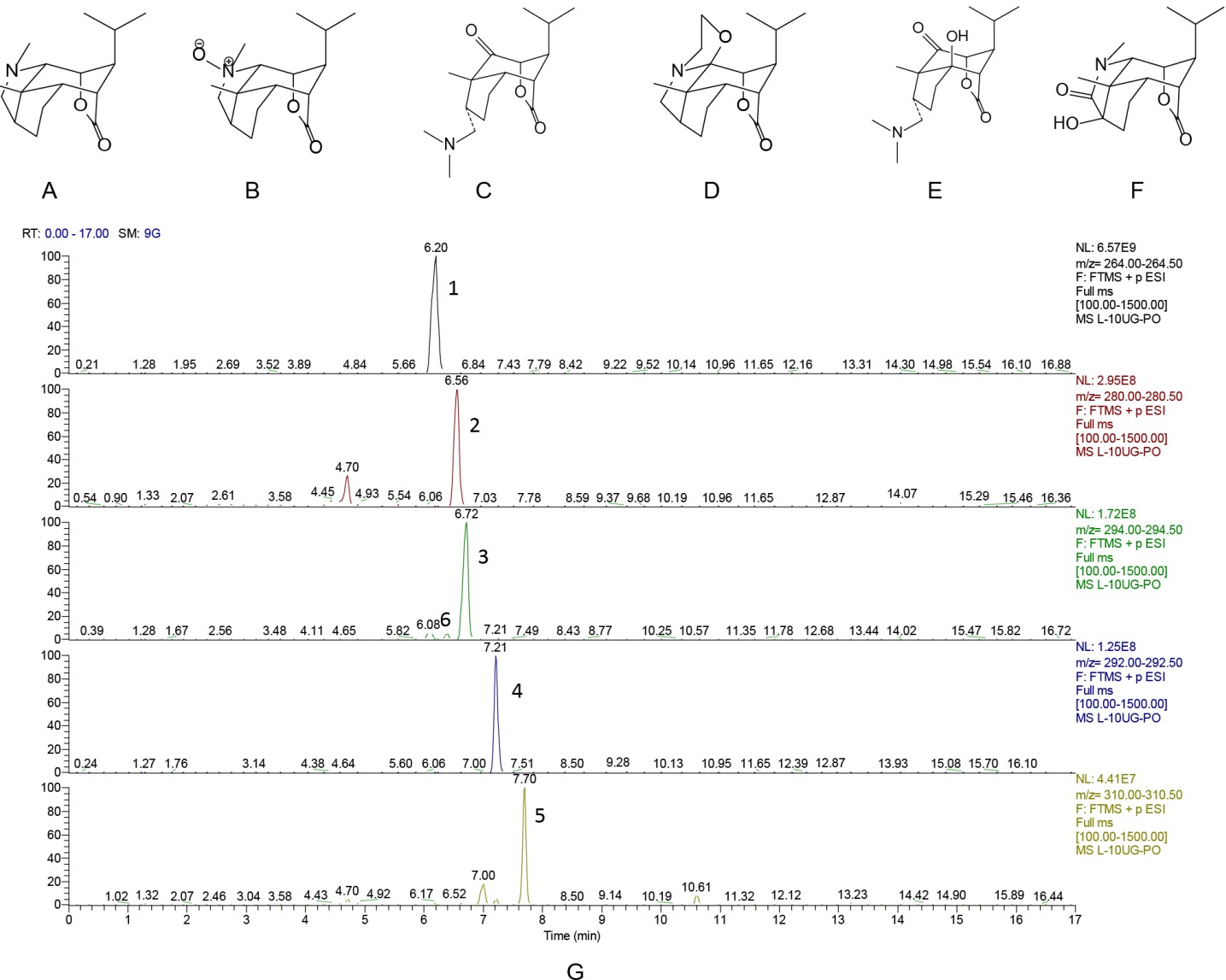

**Figure 1 Chemical structures and chromatograms of Dendrobium alkaloids.** Chemical structures and chromatograms of Dendrobium alkaloids (A) Dendrobine, (B) Dendrobine-N-oxide, (C) Nobilonine, (D) Dendroxine, (E) 6-Hydroxy-nobilonine, and (F) 13-Hydroxy-14-oxodendrobine. (G) Chromatogams of Dendrobium alkaloids. Differences between the accurately tested and calculated m/z values were <2 ppm for [M+H]+ions: (1) Dendrobine, (2) Dendrobine-N-oxide, (3) Nobilonine, (4) Dendroxine, (5) 6-Hydroxy-nobilonine, and (6) 13-Hydroxy-14-oxodendrobine.

3.3%), Nobilonine ($C_{17}H_{27}O_3N$, 2.0%), Dendroxine ($C_{17}H_{25}O_3N$, 0.9%), 6-Hydroxy-nobilonine ($C_{17}H_{27}O_4N$, 0.32%), and 13-Hydroxy-14-oxodendrobine ($C_{16}H_{23}O_4N$, 0.07%). The chemical structures of these ingredients were shown in the Fig. 1A, and the chromatograms of the sample solutions were shown in the Fig. 1B.

$A\beta_{25-35}$ (Sigma, St Louis, MO, USA) was dissolved in physiological saline to a final concentration of 2.5 g/L, and incubated for one week at 37 °C to reach a state of aggregation. CNTF antibodies (ab190985) and GDNF antibodies (ab18956) were obtained from Abcam (Cambridge, England). BDNF (sc-546) was purchased from Santa Cruz Biotechnologies

(Santa Cruz, CA, USA). TUNEL kits (*In Situ* Cell Death Detection Kit and POD) were purchased from Roche (Switzerland).

## Animals

Male Kunming mice (20–25 g) were purchased from the Laboratory Animal Center, Chongqing, China (Grade: specific pathogen-free [SPF], Certificate no.: SCXK 2012-0005). Mice were housed in SPF-grade animal facilities (Certificate no.: SYXK 2014-003), with 22–23 °C, and a 12-hour light/dark cycle. Mice were given food and water freely. All animal procedures were approved by the animal experimental ethical committee of Zunyi Medical College.

## Experimental designs

The animals were randomly divided into three groups ($n = 6$): sham surgery, model, and DNLA (40 mg/kg) treatment group. Mice were anesthetized with 7% chloral hydrate (35–45 mg/kg, *i.p.*) and fixed on a stereotaxic instrument (RWD Life Science, Guangdong, China). Then A$\beta_{25-35}$ (2.5 µg/µL, 2 µL) was injected into both lateral cerebral ventricle in the model and DNLA groups *via* a 5-µl microinjector. The injection site was posterior from the bregma (AP) $= -0.4$, mediolateral from the midline (MR) $= 1.2$, and dorsoventral from the skull (DV) $= 2.7$. Mice in the sham group were treated the same as the model group except for the injection of A$\beta_{25-35}$ instead of sterile normal saline (*Nie et al., 2010*). Mice in the DNLA group were administered intragastrically with DNLA (40 mg/kg) on the first day after the injection for 19 consecutive days, while mice in the sham and model groups were administered with distilled water.

## Morris water maze test

The ability of spatial learning and memory of the mice was evaluated through the Morris Water Maze (MWM) test (*Edwards et al., 2014*; *Milner et al., 2014*). The test was performed on the 14th day after drug administration. The pool of MWM was 120 cm in diameter, filled with water (22–25 °C) and divided into four quadrants. A hidden platform was placed in the center of the target quadrant 1.5–2.0 cm under the surface of the water. MWM included the place navigation test and the spatial probe test. In the place navigation test, all mice were trained three times per day for four days. Mice were initially placed in the three quadrants which did not have a platform. Each trail lasted for 60 s or ended as soon as the mouse climbed on the platform. The time was recorded as the escape latency, if the mouse climbed on the platform within 60 s. If the mouse failed to find the platform within 60 s, its escape latency was recorded as 60 s. The spatial probe test was performed following the end of the place navigation test. In this test, the platform was removed, and each mouse was allowed to swim for 60 s. The searching distance in the target area and total area were measured. Data was recorded and analyzed by Top View Animal Behavior Analyzing System (Version 3.00).

## Morphometric analysis

After the MWM test, three mice were selected randomly from each group and anesthetized with 7% chloral hydrate. Then these mice were perfused with phosphate-buffered saline

(0.1 M, 4 °C) *via* the ascending aorta, followed by 4% paraformaldehyde until the tail and limbs were rigid. Thereafter, the brains were removed and cut in half. Half of the brain was fixed with 4% paraformaldehyde for seven days and cut into coronal sections (4-μm thick) for hematoxylin-eosin (H&E), Nissl and TUNEL staining (*Yang et al., 2014*; *Liu et al., 2015*). The other half was fixed with 2.5% glutaraldehyde for electron microscope detection.

## Western blot analysis

Western blot analysis was performed as previously reported (*Liu et al., 2015*; *Li et al., 2015*). Three mice from each group were sacrificed after WMW test, and the hippocampal and cortical tissues were collected and homogenized in RIPA lysis buffer (1:5, w/v). Protein concentrations were determined by BCA protein assay kit. An aliquot of 45 μg of protein was applied for electrophoresis followed by the transferring of protein to the polyvinylidene difluoride (PVDF) membranes. The membranes were blocked with 5% nonfat dry milk in TBST buffer for one hour at room temperature, and incubated with the primary antibody: anti-CNTF (1:1,000), BDNF (1:1,000), and GDNF (1:1,000) at 4 °C overnight. Next, HRP-labeled goat anti-rabbit IgG (Beyotime Biotechnology, Jiangsu, China; A0208, 1:2,000) was incubated with the membrane at room temperature for one hour. The blots were visualized using the enhanced ECL Western blot detection kit (7Sea Biotech, Shanghai, China) and scanned to Gel Imaging. The band intensity was quantified using Quantity One 1-D analysis software v4.52 (BioRad, Hercules, California, USA).

## Statistical analysis

All data were presented as mean $\pm$ standard error. Data were analyzed by SPSS 13.0 statistics software by one-way ANOVA or student's $t$-test. $P < 0.05$ was considered as statistically significant.

## RESULTS

### Protective effects of DNLA on learning and memory deficits induced by A$\beta_{25-35}$

Compared with the sham group, the escape latency of the model group was significantly prolonged in the navigation test, and the explored distance in the target area was decreased in the spatial probe test. After treatment for 14 days, the escape latency of the DNLA group was decreased and the explored distance in the target area was increased as compared with the model group. The results indicated that DNLA could protect the mice against learning and memory deficits which were induced by A$\beta_{25-35}$ injection (Fig. 2).

### Morphological changes of cortex and hippocampus tissues

H&E staining was performed to observe the morphological changes of cortex and hippocampus tissues in mice brain. The results showed that the morphology of brain tissues in the sham group was normal, and the cell density was higher. However, loose structures in hippocampal and cortical neurons with disordered hippocampal pyramidal cell layers occurred in the model group. Furthermore, the images showed that not only the number of neurons was reduced, but also that neurons staining were abnormal with

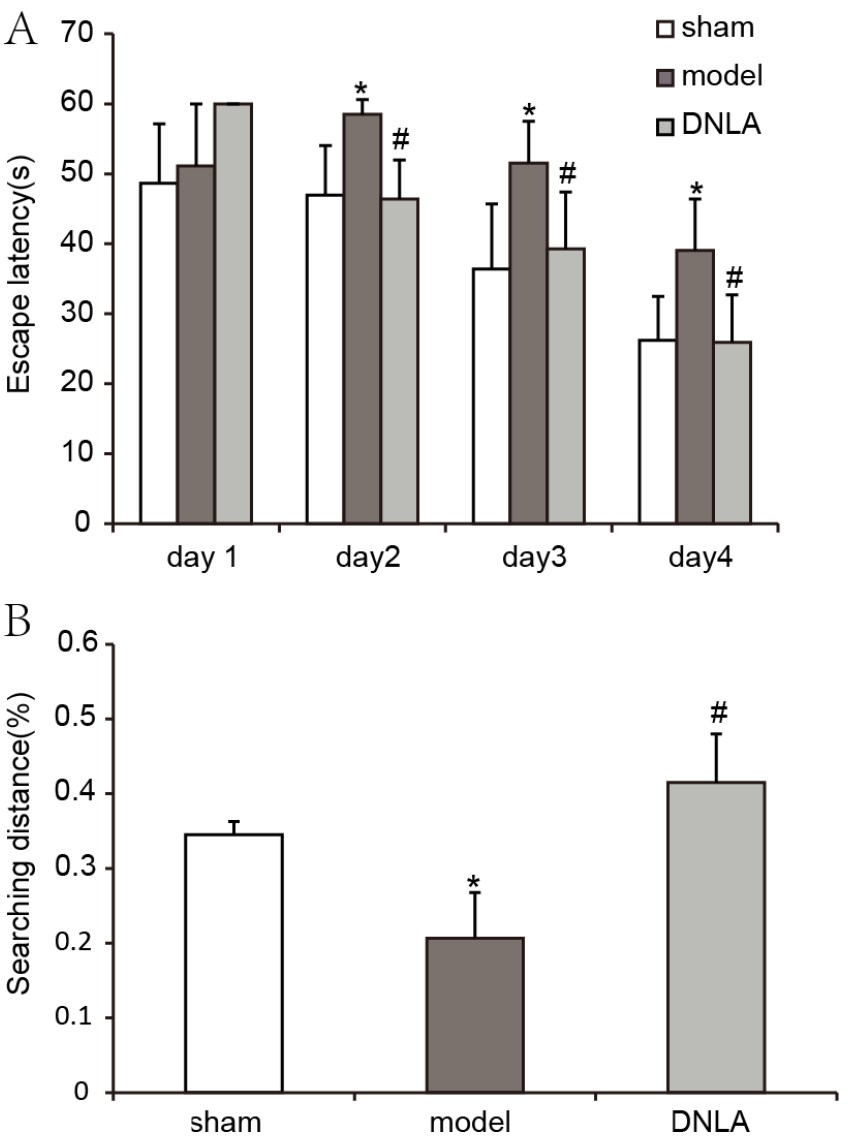

**Figure 2  Results of the Morris water maze test.** Effect of DNLA on A$\beta_{25-35}$-induced learning and memory impairment. (A) The graph represented the escape latency of the different groups. (B) Graphs described the adjusted searching distance in the space probe test. Data were expressed as mean $\pm$ SEM ($n = 6$). $^*P < 0.05$ *vs.* the sham group; $^\#P < 0.05$ *vs.* model.

nuclear condensation in the model group. DNLA treatment significantly decreased the number of abnormally stained neurons. Also, the morphology of brain tissues was generally normal in the DNLA group (Fig. 3).

## Effect of DNLA on apoptosis in the cortex and hippocampus

The effect of DNLA on neuronal apoptosis induced by A$\beta_{25-35}$ was observed using the TUNEL method. The results revealed that the number of apoptotic cells in the hippocampus and cortex were not affected in the sham group, but were increased in the model group. DNLA treatment decreased the number of apoptotic cells (Fig. 4).
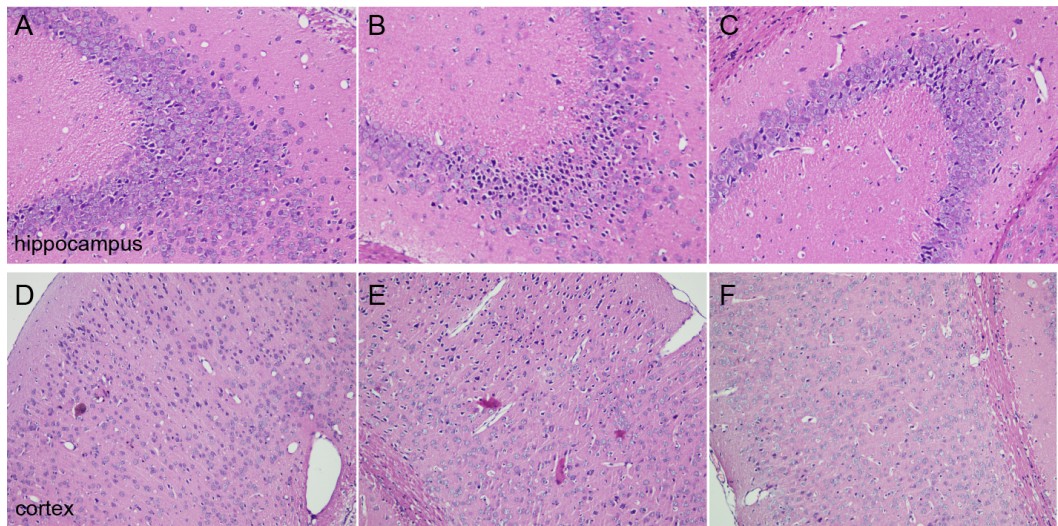

**Figure 3  Images of H&E staining.** Effects of DNLA on morphological alterations in the hippocampus and cortex induced by $A\beta_{25-35}$. Sections of the hippocampus CA3 region and cortex were obtained and stained with H&E (magnification, 200×). (A–C) showed the CA3 hippocampal cyto-architecture of mice in sham, model and DNLA groups, respectively; (D–F) showed the cortical cyto-architecture of mice in the sham group, model group and DNLA groups, respectively.

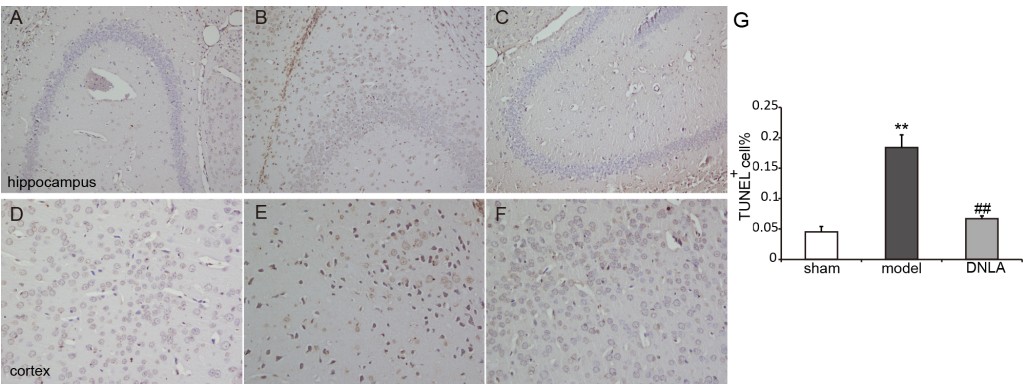

**Figure 4  Images of TUNEL staining.** Effects of DNLA on A$\beta$25-35-induced neuronal apoptosis in hippocampus and cortex. Sections of the hippocampus CA3 region and cortex were obtained and stained with TUNEL 184 staining (magnification, 200×). (A–C) showed the CA3 hippocampal cyto-architecture of mice in sham, model and DNLA groups, respectively; (D–F) showed the cortical cyto-architecture of mice in the sham group, model group and DNLA groups, respectively. The cells dyed brown are apoptotic cells. (G) The number of the apoptotic neurons in hippocampus and cortex ($\bar{x} \pm SEM, n = 3$). **$P < 0.01$ vs. the sham group; ##$P < 0.01$ vs. the model group.

## Changes in Nissl bodies in cortex and hippocampus tissues

As a neural characteristic structure, the number of Nissl bodies reflects the state of neurons. In physiological conditions, the Nissl bodies were big and abundant, showing that the function of neuronal protein synthesis was strong; on the other hand, when nerve cells were damaged, the number of Nissl bodies will be reduced or even disappear. Intraneural Nissl bodies of the cortex and hippocampus were lightly stained and appeared to be sparsely
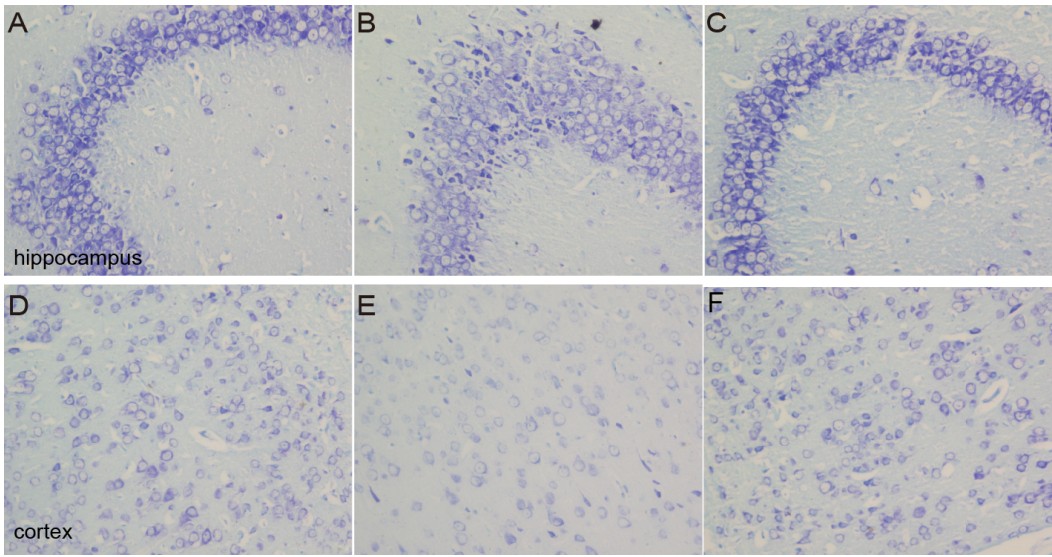

**Figure 5  Images of Nissl staining.** Effects of DNLA on A$\beta_{25-35}$-induced neuronal Nissl bodies in the hippocampus and cortex. Sections of the hippocampus CA3 region and cortex were obtained and stained with Nissl staining (400×). (A–C) showed the CA3 hippocampal cyto-architecture of mice in sham, model and DNLA groups, respectively; (D–F) showed the cortical cyto-architecture of mice in the sham group, model group and DNLA groups, respectively.

arranged in the model group. However, deeper stained Nissl bodies with higher density in cortex and hippocampus neurons were found in the sham and DNLA groups (Fig. 5).

## Effects of DNLA on the ultrastructures of neurons in A$\beta_{25-35}$-injected mice

The ultrastructures of neurons were observed under electron microscope. The results revealed that the nuclear morphology of mice in the sham group was normal, and the chromatin was normally distributed. In the cytoplasm, mitochondrial morphology was normal with clear and regularly-arranged cristae. Furthermore, a number of ribosomes could be observed in these regularly-arranged rough endoplasmic reticulum. However, in the model group, the ultrastructures of neurons were abnormal. It could be found that nuclear membrane had folded or had obscure boundaries, mitochondria were swelling even without the cristae, and the endoplasmic reticulum was swelling. DNLA treatment could reduce the injury. Compared with the model group, nuclear membrane was more smooth, swelling of mitochondrial was reduced, and partial rupture of cristae were observed in the DNLA group (Fig. 6). The neuronal synapses were also observed under electron microscope. Injection of A$\beta_{25-35}$ would reduce the number of synapses and synaptic vesicles, and cause the shape and size of synaptic vesicles uneven. The structure of synapses was obscured, and the structure of the post-synaptic lattice became thin. DNLA inhibited the loss of synapses significantly ($P < 0.05$). Also, in the DNLA group, the generally normal synaptic structures were maintained (Fig. 7).

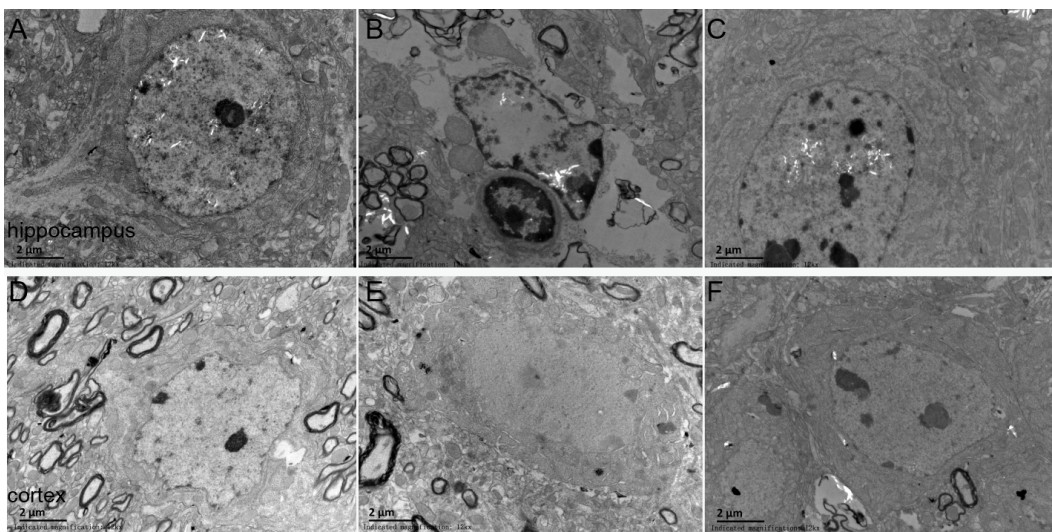

**Figure 6** **Images of the ultrastructures of neurons.** Effects of DNLA on the ultrastructures of neurons in $A\beta_{25-35}$-injected mice. The ultrastructures were observed under electron microscope (12kx ×). (A–C) showed the ultrastructures of hippocampal neurons in sham, model and DNLA groups, respectively; (D–F) showed the ultrastructures of cortical neurons in sham, model and DNLA groups, respectively.

## Effects of DNLA on the protein expressions of BDNF, CNTF and GDNF in the cortex and hippocampus

The protein expressions were analyzed by Western blot. Compared with the sham group, the expressions of BDNF, CNTF and GDNF significantly decreased in the model group. DNLA group had significantly increased expressions of BDNF, CNTF and GDNF in the cortex and hippocamous (Fig. 8).

## DISCUSSION

Numerous literatures have reported that intra-hippocampal or intra-cerebroventricular injections of $A\beta_{1-42}$ or $A\beta_{25-35}$ fragments into rats or mice could induce neuronal death, and alter spatial learning and memory performance. $A\beta$ could induce a variety of injuries, such as oxidative injury and disturbed neuronal ionic homeostasis, which could eventually result in neuronal dysfunction and selective neuronal loss (*Lue et al., 1999*; *Niikura et al., 2002*). Excessive neuronal loss can damage brain functions, and the cognitive ability is destroyed firstly. In this study, we established the mice model which was induced by the intra-cerebroventricular injection of $A\beta_{25-35}$, and found that there were a lot of apoptotic neurons in the hippocampus CA3 region and cortex. The results of MWM test indicated that neuronal apoptosis was accompanied with an obvious failure in spatial learning and memory performance. On the other hand, DNLA treatment could significantly improve the ability of spatial learning and memory, and decreased apoptosis in TUNEL and H&E staining. This result demonstrated that DNLA could improve learning and memory deficits in AD model mice, and the protection may be due to the decreased apoptosis induced by $A\beta_{25-35}$.

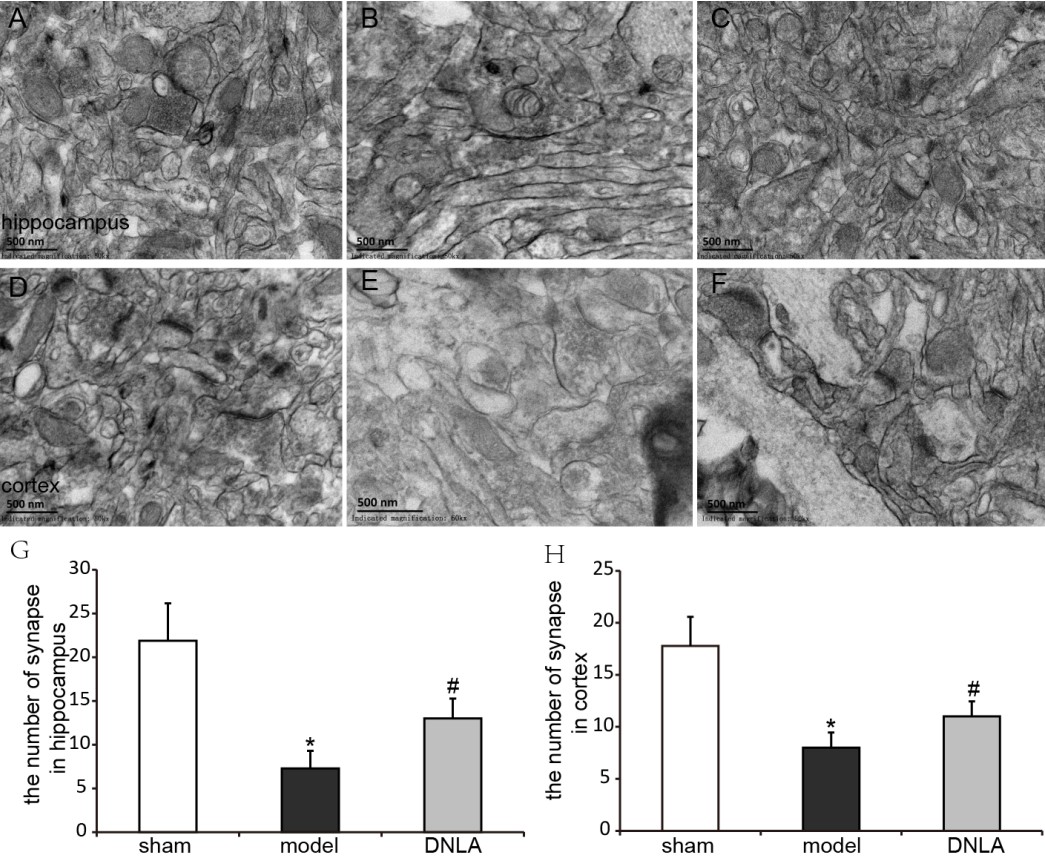

**Figure 7** **Effects of DNLA on neuronal synapses in A$\beta_{25-35}$-injected mice.** The neuronal synapses were observed under electron microscope (50kx ×). (A–C) showed the synapses of hippocampal neurons in sham, model and DNLA groups, respectively; (D–F) showed the synapses of cortical neurons in the sham, model and DNLA groups, respectively. (G) showed the effects of DNLA on the number of neuronal synapses in A$\beta_{25-35}$-induced mice. Data were expressed as mean ± SEM ($n = 3$). *$P < 0.05$ *vs.* the sham group; #$P < 0.05$ *vs.* the model group.

As it is known, when neurons are damaged, Nissl bodies are most sensitive. The main changes of Nissl bodies associated with neuronal injury include the dissolution and disappearance of Nissl bodies. In the model group, the number of Nissl bodies was decreased in cells. After the administration of DNLA, the number was increased. The results indicated DNLA could alleviate neuronal damage.

It has reported that memory disorders in AD patients begin with subtle changes in hippocampal synaptic efficacy earlier than extensive neuronal degeneration. *Davies et al. (1987)* performed a quantitative morphometric analysis to measure the densities of neurons and synapses in cerebral cortical biopsy tissues collected from AD patients. They found that the numerical density of synapses decreased 25%–35%, and the number of synapses per cortical neuron decreased 15%–35% in AD patients (*Davies et al., 1987*). Evidence also suggested that synaptic dysfunction is caused by oligomeric assemblies of A$\beta$ (*Lacor et al., 2004*; *Rowan et al., 2007*). Even in very mildly impaired patients, the degree of synaptic loss in the cortex showed a significant association with soluble A$\beta$ levels. Our results

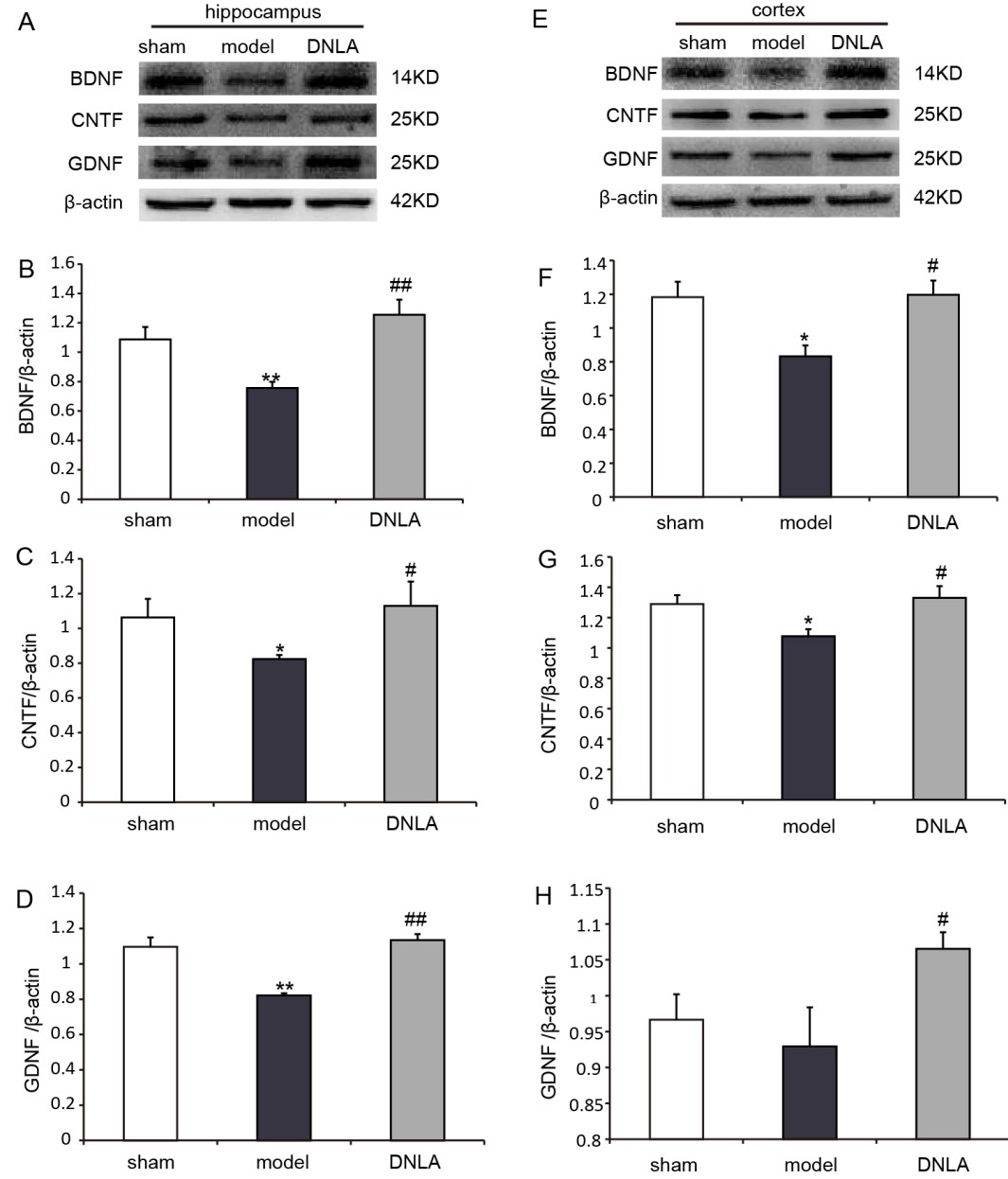

**Figure 8  Results of western blot.** Effects of DNLA on the protein expressions of BDNF, CNTF and GDNF in brain tissues. (A) Protein contents of hippocampal tissues were plotted for sham, model and DNLA groups, respectively. The corresponding quantitation of BDNF, CNTF, GDNF protein were showed in (B–D) respectively. (E) Protein contents of cortex tissues were plotted for sham, model and DNLA groups, respectively. The corresponding quantitation of BDNF, CNTF, GDNF protein were showed in (F–H) respectively. The relative optical density was normalized to $\beta$-actin. Data were expressed as mean $\pm$ SEM ($n = 3$). Significance $^*P < 0.05$, $^{**}P < 0.01$ compared to the sham group, $^{\#}P < 0.05$, $^{\#\#}P < 0.01$ compared to the model group.

revealed that A$\beta_{25-35}$ induced synapse loss in the hippocampus and cortex, and DLNA could dramatically inhibit the loss of synapses and improve synaptic structures, thereby improving the learning and memory abilities of mice.

The application of neurotrophic factors could enable the modulation of neuronal survival and synaptic connectivity (*Poo, 2001*). Neurotrophic factors are synthesized and secreted by the target tissue. After binding with receptors (tyrosine receptor kinase [Trk] and pan-neurotrophin receptot [p75]), these factors are internalized and transported to the cell body in a retrograde manner, where they would affect neuronal survival and differentiation (*Poo, 2001*; *Barker & Shooter, 1994*). The loss of neurons in the AD process is at least partly due to the lack of one or more neurotrophic factors. It has been found that the mRNA expression level of BDNF is low in the hippocampus and cortex of AD patients. *Christensen et al. (2008)* reported that the injection of A$\beta_{1-42}$ in the hippocampus of rat would lead to learning and memory impairment and reduce cortical BDNF levels. These results indicate that memory deficit induced by A$\beta_{1-42}$ is associated with the disorder of the expression of BDNF, which is reflected in lower cortical BDNF levels. Another study has shown that secretory vesicles containing BDNF exist within axon terminals, dendrites of pyramidal, and granule cells. BDNF can be secreted into the synaptic gap and binds with TrkB. After binding, BDNF serves as a key regulator in synaptic plasticity and memory, and plays a role in long-term and short-term memory (*Lu, Christian & Lu, 2008*). The present study proved that intracerebroventricular injection of A$\beta_{25-35}$ could decrease the expression of BDNF in the hippocampus and cortex. However, DNLA have significantly increased BDNF expression, suggesting that the abilities of DNLA to reduce neuronal apoptosis and synaptic loss in mice may be partly associated with increased BDNF protein level.

The maintenance of the neuronal survival and function needs simultaneous support from a variety of neural survival factors, which form an information network through signal transduction pathways in nerve cells, and generate an amplified effect *via* intracellular generalization (*Poo, 2001*). Therefore, we examined two other neurotrophic factors, CNTF and GDNF. CNTF is a member of a cytokine family (*Duff & Baile, 2003*), and is known for its neuroprotective effects, being as a survival factor for sympathetic, sensory and hippocampal neurons. Garcia et al. reported that CNTF reduced the impairments of synaptic and cognitive functions in the Tg2576 AD mouse model, and rescues neurons from degeneration induced by A$\beta$ in vitro and  in vivo (*Pasquin, Sharma & Gauchat, 2015*; *Garcia et al., 2010*). GDNF family and its different receptor systems are now recognized as one of the major neurotrophic networks in the nervous system, which are important for the development, maintenance and function of neurons and glial cells. GDNF signaling through NCAM can result in the activation of kinases MAP and Src-like kinases, and contributes to the regulation of several different processes, including synapse formation and neurite outgrowth (*Allen et al., 2013*). In hippocampal neuronal cultures, GDNF has been found to increase the number of synapses. Although most studies on the function of GDNF and its receptors has been concentrated on midbrain dopaminergic neurons, the receptors of GDNF are observed in many other brain regions. A study analyzed 250 blood plasma samples of AD patients, and identified GDNF as one of 18 signaling

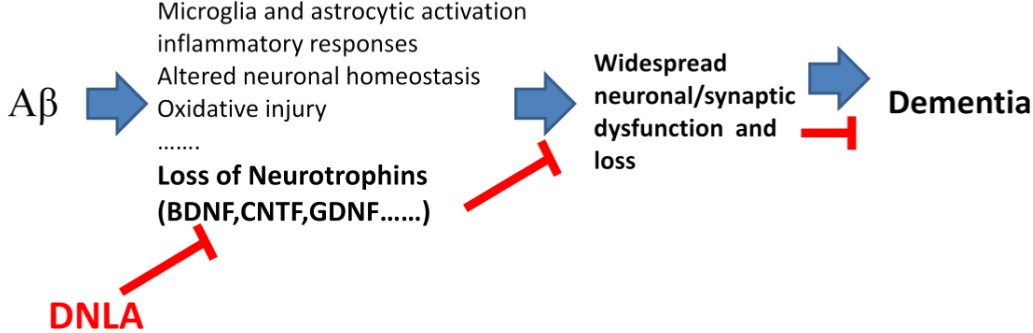

**Figure 9 Overview diagram illustrating the neuronal damage induced by A$\beta_{25-35}$ and DNLA intervention.** DNLA could prevent neuronal apoptosis and synaptic loss *via* increasing the expression of BDNF, GDNF and CNTF in the hippocampus and cortex, improving A$\beta$-induced spatial learning and memory impairment in mice.

proteins. GDNF level was lower in AD patients than the controls. It is not difficult to envision how a signaling molecule such as GDNF, which has potent effects on neuronal maturation, cell survival and synapse formation, can effect cognitive functions; and if aberrantly or insufficiently expressed or secreted, contribute to cognitive decline (*Ibáñez & Andressoo, 2016*; *Pertusa et al., 2008*). Our study also found DNLA can significantly increase the expression of CNTF and GDNF. The results indicated that the ability of DNLA to prevent neuronal and synaptic loss may be attributed to increase the expression of BDNF, CNTF and GDNF.

In summary, the present study demonstrates that DNLA could improve A$\beta$-induced learning and memory impairment in mice. This may be due to the prevention of neuronal apoptosis and synaptic loss via the increased expression of BDNF, GDNF and CNTF in the hippocampus and cortex (Fig. 9).

### Funding

This work has been supported by the National Natural Science Foundation of China (Grant No. 81473201, 81660685) and Science Research Project of Education Department in Guizhou Province (NO. KY[2015]373). The funders had no role in study design, data collection and analysis, decision to publish, or preparation of the manuscript.

### Grant Disclosures

The following grant information was disclosed by the authors:
National Natural Science Foundation of China: 81473201, 81660685.
Science Research Project of Education Department in Guizhou Province: KY[2015]373.

### Competing Interests

The authors declare there are no competing interests.

## Author Contributions

- Jing Nie conceived and designed the experiments, performed the experiments, analyzed the data, contributed reagents/materials/analysis tools, wrote the paper, prepared figures and/or tables.
- Yong Tian and Yu Zhang performed the experiments.
- Yan-Liu Lu analyzed the data, contributed reagents/materials/analysis tools.
- Li-Sheng Li contributed reagents/materials/analysis tools.
- Jing-Shan Shi conceived and designed the experiments, reviewed drafts of the paper.

## Animal Ethics

The following information was supplied relating to ethical approvals (i.e., approving body and any reference numbers):

Animal Ethics Committee of Zunyi Medical College.

## Field Study Permissions

The following information was supplied relating to field study approvals (i.e., approving body and any reference numbers):

Dendrobium was collected from Dendrobium planting regions of Xintian Traditional Chinese Medicine Industry Development Co., Ltd of Guizhou Province.

## Data Availability

The raw data has been supplied as a Supplementary File.

## Supplemental Information

Supplemental information for this article can be found online at http://dx.doi.org/10.7717/peerj.2739#supplemental-information.

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
