# Peer review of "Dendrobium alkaloids prevent Aβ25–35-induced neuronal and synaptic loss via promoting neurotrophic factors expression in mice"

_PeerJ, doi:10.7717/peerj.2739_

## Round 0.1 · original submission · Major Revisions

All the reviewers found your manuscript is of merit, and is interesting to the readers of PeerJ. However, the reviewers raised a number of questions which need to be carefully addressed point by point to improve your papers before it can be accepted for publication.

Reviewer 1 ·

Basic reporting

see below

Experimental design

see below

Validity of the findings

see below

Additional comments

The authors have investigated the protective effects of Dendrobium alkaloids on beta-amyloid peptide segment 25-35-induced neuron and synaptic loss in mice. The study was well designed and carefully carried out. Data presented strongly support the central hypothesis. The manuscript was presented in a logical and scientific way. The authors are encouraged to revise the following aspects to improve their manuscript.
1. There is ambiguity in the title. The authors are encouraged to change it into “Dendrobium alkaloids prevent Ab25-35-induced neuronal and synaptic loss via promoting neurotrophic factors expression in mice”.
2. Fig. 1B needs to be enlarged, because the fonts are barely readable.
3. A graphic diagram of all players contributing to the protective effects of DNLA in Ab-induced neuron and synaptic loss in mice model.

Reviewer 2 ·

Basic reporting

The manuscript is well-written, has important messages, and should be of great interest to the readers. However, there are still some issues need to be addressed. Listed below are specific comments.

Experimental design

see below

Validity of the findings

see below

Additional comments

The authors demonstrated that DNLA is effective in preventing against neuronal and synaptic loss via promoting neurotrophic factors expression induced by Aβ25-35 in mice. The manuscript is well-written in general, has important messages, and could be of great interest to the readers. However, there are still some issues need to be addressed. In general, I have a few comments and suggestions to the authors highlighted below.

1. DNLA decreased apoptosis and promoted neurotrophic factors which may be important in the prevention of neuronal and synaptic loss. The authors are encouraged to provide other possible mechanisms in their discussion.

2. Please indicate the reasons for the dose of DNLA 40mg/kg. Is there any dosage curve selection? It is better to have a positive control in this manuscript

3. Fig4b effects of DNLA on Aβ25-35-induced neuronal apoptosis in brain tissues. Tissues should be clearly indicated for hippocampus or cortex, or both of them.

Reviewer 3 ·

Basic reporting

This is an interesting topic linked the potential neuronal protective roles of Dendrobium alkaloids to the Alzheimer's disease. The authors provided lines of significant evidence that the Dendrobium alkaloids could prevent neuron and synaptic damages in Aβ25-35-induced Alzheimer's disease model. Furthermore the western blots experiments indicated that the Dendrobium might have the protective roles by increasing the expression of some neurotrophic factors such as BDNF, CNTF and GDNF. However, several places need to be addressed more in detail and clearly.

Experimental design

For materials and methods, please provide more detailed information and carefully not missing important messages. For example, there is no description about DNLA group in the 2.4 Surgery section. Did you inject the same amount of Aβ25-35? Did you apply the DNLA at the same time or the same day when you injected the Aβ25-35? Actually, the 2.3 and 2.4 can be combined in experimental designs.

Validity of the findings

First of all, based on the experiment designs (“Aβ25-35(10 μg) was injected into the bilateral ventricles of male mice followed by an oral administration of DNLA (40 mg/kg) for 19 days“), It is hard to figure out that the protective roles of Dendrobium were due to less neuronal injury or due to the neuronal cell recovery. The results showed that the Dendrobum treated group has less neuronal injury than model group compared to sham group. But some the expressions are confused when mixing comparison of model to sham, and then Dendrobium-treated to model. For example, it stated that the DNLA increased the number of Nissl bodies and synapses. How could you say DNLA increased them not inhibited the loss of them? Therefore, a DNLA-only treatment group would be added in to get further conclusion.
Please provide more detailed information about why you chose Aβ25-35 not Aβ1-42 as model and why you applied 40mg/kg DNLA.

Additional comments

This is an interesting topic linked the potential neuronal protective roles of Dendrobium alkaloids to the Alzheimer's disease. The authors provided lines of significant evidence that the Dendrobium alkaloids could prevent neuron and synaptic damages in Aβ25-35-induced Alzheimer's disease model. Furthermore the western blots experiments indicated that the Dendrobium might have the protective roles by increasing the expression of some neurotrophic factors such as BDNF, CNTF and GDNF. However, several places need to be addressed more in detail and clearly.
First of all, based on the experiment designs (“Aβ25-35(10 μg) was injected into the bilateral ventricles of male mice followed by an oral administration of DNLA (40 mg/kg) for 19 days“), It is hard to figure out that the protective roles of Dendrobium were due to less neuronal injury or due to the neuronal cell recovery. The results showed that the Dendrobum treated group has less neuronal injury than model group compared to sham group. But some the expressions are confused when mixing comparison of model to sham, and then Dendrobium-treated to model. For example, it stated that the DNLA increased the number of Nissl bodies and synapses. How could you say DNLA increased them not inhibited the loss of them? Therefore, a DNLA-only treatment group would be added in to get further conclusion.
Please provide more detailed information about why you chose Aβ25-35 not Aβ1-42 as model and why you applied 40mg/kg DNLA.
For materials and methods, please provide more detailed information and carefully not missing important messages. For example, there is no description about DNLA group in the 2.4 Surgery section. Did you inject the same amount of Aβ25-35? Did you apply the DNLA at the same time or the same day when you injected the Aβ25-35? Actually, the 2.3 and 2.4 can be combined in experimental designs.
In discussion part, some contents such as AD are redundant as stated in introduction. Please remove or summarize it.
Labels on figures were too hard to see. Please improve the contrast and fond size. Figure legends need more clear description.
Others, there were many typos or expressions needed to be revised. Listed below are not all, just for references.
1. Abstract- Conclusions: “at least”
2. Line 54, “an a”; Line 54, should start a new sentence, use“They”
3. Line 62, “improve neuronal damage”? “prevent neuronal damage” would be better.
4. Line 105, 5μ g/2μ l ?
5. Line 147, “data was” change to “data were”
6. Line 191, “when never”?

---

## Round 0.2 · accepted · Accept

We are looking forward to receiving your novel contributions in the future.

Reviewer 1 ·

Basic reporting

see below

Experimental design

see below

Validity of the findings

see below

Additional comments

The authors have investigated the protective effects of Dendrobium alkaloids on beta-amyloid peptide segment 25-35-induced neuron and synaptic loss in mice. The study was well designed and carefully carried out. Data presented strongly support the central hypothesis. The manuscript was presented in a logical and scientific way. The authors have addressed all questions and the reviewer does not have further suggestion. The manuscript is in good shape to be published.

Reviewer 3 ·

Basic reporting

This manuscript has been revised based on the comments of the previous review work and It meets the requirement for publication. If the letters labeled on the figures (Figures 6 and 7) could be more visible, that would be great. You may consider put the "A, B, C" outside the figures so it is easier to edit for the "black/white"background figures like these.

Experimental design

It is revised appropriately.

Validity of the findings

It is revised appropriately.

Additional comments

This manuscript has been revised based on the comments of the previous review work and It meets the requirement for publication. If the letters labeled on the figures (Figures 6 and 7) could be more visible, that would be great. You may consider put the "A, B, C" outside the figures so it is easier to edit for the "black/white"background figures like these. Thanks for your efforts to the revision of the manuscript.